Can phosphorus application and cover cropping alter arbuscular mycorrhizal fungal communities and soybean performance after a five-year phosphorus-unfertilized crop rotational system?

http://orcid.org/0000-0002-9314-6994 Higo Masao higo.masao@nihon-u.ac.jp
Sato Ryohei
Serizawa Ayu
Takahashi Yuichi
Gunji Kento
Tatewaki Yuya
Isobe Katsunori
Department of Agricultural Bioscience, College of Bioresource Sciences, Nihon University , Kanagawa , Japan
Maloof Julin
Electronic publication date: 2018 Apr 18
Publication date: 2018
Volume: 6
Electronic Location ID: e4606
Received 2017 Jun 1; Accepted 2018 Mar 22
Copyright: © 2018 Higo et al.
Copyright year: 2018
Copyright holder: Higo et al.
License: This is an open access article distributed under the terms of the Creative Commons Attribution License, which permits unrestricted use, distribution, reproduction and adaptation in any medium and for any purpose provided that it is properly attributed. For attribution, the original author(s), title, publication source (PeerJ) and either DOI or URL of the article must be cited.
License URL: https://creativecommons.org/licenses/by/4.0/

Keywords: Arbuscular mycorrhizal fungi, Community structure, Cover cropping, Phosphorus fertilization, Soybean

Funding: Nihon University College of Bioresource Sciences Research 14C-001 and 15C-001 This study was financially supported by a grant from the Nihon University College of Bioresource Sciences Research Grant (14C-001 and 15C-001). The funders had no role in study design, data collection and analysis, decision to publish, or preparation of the manuscript.

==============================
Background

Understanding diversity of arbuscular mycorrhizal fungi (AMF) is important for optimizing their role for phosphorus (P) nutrition of soybeans (Glycine max (L.) Merr.) in P-limited soils. However, it is not clear how soybean growth and P nutrition is related to AMF colonization and diversity of AMF communities in a continuous P-unfertilized cover cropping system. Thus, we investigated the impact of P-application and cover cropping on the interaction among AMF colonization, AMF diversity in soybean roots, soybean growth and P nutrition under a five-year P-unfertilized crop rotation.

Methods

In this study, we established three cover crop systems (wheat, red clover and oilseed rape) or bare fallow in rotation with soybean. The P-application rates before the seeding of soybeans were 52.5 and 157.5 kg ha−1 in 2014 and 2015, respectively. We measured AMF colonization in soybean roots, soybean growth parameters such as aboveground plant biomass, P uptake at the flowering stage and grain yields at the maturity stage in both years. AMF community structure in soybean roots was characterized by specific amplification of small subunit rDNA.

Results

The increase in the root colonization at the flowering stage was small as a result of P-application. Cover cropping did not affect the aboveground biomass and P uptake of soybean in both years, but the P-application had positive effects on the soybean performance such as plant P uptake, biomass and grain yield in 2015. AMF communities colonizing soybean roots were also significantly influenced by P-application throughout the two years. Moreover, the diversity of AMF communities in roots was significantly influenced by P-application and cover cropping in both years, and was positively correlated with the soybean biomass, P uptake and grain yield throughout the two years.

Discussion

Our results indicated that P-application rather than cover cropping may be a key factor for improving soybean growth performance with respect to AMF diversity in P-limited cover cropping systems. Additionally, AMF diversity in roots can potentially contribute to soybean P nutrition even in the P-fertilized cover crop rotational system. Therefore, further investigation into the interaction of AMF diversity, P-application and cover cropping is required for the development of more effective P management practices on soybean growth performance.

Introduction

Phosphorus (P) application of synthetic fertilizers is often required to achieve high productivity due to strong interactions of P with soil compounds such as iron and aluminum (Lynch, 2007). Additionally, increased application of synthetic fertilizers can lead to less active microbial-mediated processes of mineralization and solubilization, and increase the potential risk of environmental pollution (Bai et al., 2013). The expected global peak in P production is also predicted to occur around 2,030, whereas P demand is projected to increase (Cordell, Drangert & White, 2009). Moreover, current global P reserves may be depleted in 50–100 years’ time (Cordell, Drangert & White, 2009). The global average cash production costs of phosphate rock in 1987 and 2017 increased from US $31 to US $99 per metric ton during this 30 year period (Global Economic Monitor Commodities, 2017: http://databank.worldbank.org). Therefore, managing soil P availability is required to maintain agricultural crop production (Mishima et al., 2003).

Arbuscular mycorrhizal fungi (AMF) can increase host plant P uptake and growth, and AMF may especially improve plant P and micronutrients uptake (Smith & Read, 2008). AMF also may act against the depletion of global P reserves (Gilbert et al., 2000). These nutritional benefits from AMF can be remarkably improved via appropriate agricultural managements (Kahiluoto et al., 2001; Kahiluoto, Ketoja & Vestberg, 2012; Gosling et al., 2006). Some species of the family Glomeraceae such as Funneliformis mosseae, Rhizophagus irregularis and family Gigasporaceae have been shown to have a positive impact on growth and nutrient uptake of plants under native and commercial AMF inoculated conditions (Verbruggen & Kiers, 2010; Gosling, Jones & Bending, 2016). Also, previous studies have shown that P uptake via AMF is a distinct functional alternative to direct uptake by plants (Bucher, 2007), and the most of the P supplied by plants can be obtained via the mycorrhizal route (Smith, Smith & Jakobsen, 2003).

According to a study by Johnson et al. (1991), there was a link between yield declines under continuous soybean cropping and the shift in AMF communities. Continuous cropping selects for the most rapidly growing and sporulating AMF species, which decreases crop performance over time. This abundance of detrimental AMF species leads to a decline in beneficial AMF species (Johnson et al., 1991). Furthermore, the introduction of mycorrhizal cover crops during winter season can be necessary for maintenance and increase indigenous AMF inoculum or diversity in soil and roots for subsequent crops (Higo et al., 2010, 2015a, 2016). Thus, the introduction of cover crops in temperate agricultural ecosystems, such as wheat, barley, oilseed rape or leguminous crops, including hairy vetch, red clover and white clover, reduces seasonal fallow and thus provides many benefits for subsequent crops and soil fertility (Karasawa & Takahashi, 2015). In addition, a diverse AMF species composition and diversity can maximize the benefits from AMF (Maherali & Klironomos, 2007; Powell et al., 2009). In turn, the diversity of AMF communities can contribute to plant P nutrition (van der Heijden et al., 1998; Verbruggen et al., 2013). Moreover, increasing AMF diversity in agroecosystems has been suggested to have the ability to boost crop growth, nutrient uptake and sustainability can be widespread (Hart & Forsythe, 2012). The diversity of AMF communities can be influenced by agricultural management practices such as land-use type (Zhao et al., 2017), crop rotation (Higo et al., 2013, 2015a), tillage (Alguacil et al., 2008), fertilization (Xiang et al., 2016) and P-application (Kahiluoto, Ketoja & Vestberg, 2009, 2012; Wang, White & Li, 2017). Recent studies have shown that the diversity of AMF communities in soils was not impacted (Jansa et al., 2014; Islas et al., 2016) or was decreased (Lin et al., 2012; Camenzind et al., 2014) by P-application. In addition, P-application may decrease (Alguacil et al., 2010; Liu et al., 2012; Gosling et al., 2013) or not impact (Beauregard et al., 2013; Liu et al., 2016) AMF diversity in roots.

The yield and growth of soybeans under a P-unfertilized four-year winter crop-soybean rotational system gradually has been found to decrease over time because of both a decrease in AMF colonization of soybeans and continuous nutrient removal from the soil by continuous crop rotations (Isobe et al., 2014). The same research group also found that there was a positive correlation between AMF colonization and soybean grain yield in a four-year consecutive winter cover crop-soybean rotational system without P fertilizer, suggesting that higher AMF colonization can be a better solution for improving soybean growth and grain yield in the P-limited soil. Cover cropping alone would also appear not to supply enough P nutrition to recover soybean performance as much as the use of an alternative way of using moderate P-application in the consecutive P-unfertilized cover crop rotational system (Karasawa & Takahashi, 2015). However, it still remains unclear which factor, such as P-application or cover cropping is driving increases in soybean performance via AMF benefits. Little is also known about how P-application and cover cropping are linked to AMF benefits and soybean growth, and the effectiveness of AMF in cover crop-soybean rotational systems to improve the reliability and the robustness of the agricultural managements.

Therefore, we hypothesized that P-application or cover cropping in a P-limited soil would increase the diversity of AMF communities and the shift would be related to the soybean growth responses in the five-year P-unfertilized cover crop study. Therefore, we approached this study with two objectives: First, to understand whether or not P-application and cover cropping impacts soybean growth performance. Second, to determine how AMF diversity in soybean roots is affected by P-application and cover cropping.

Materials and Methods

Experimental design

We conducted a field trial of winter cover crop-soybean rotation at Nihon University, in Kanagawa, Japan (35°22′N 139°27′E). The soil at the field site is classified as a volcanic ash soil (Allophonic andosol). According to the Japan Meteorological Agency (http://www.jma.go.jp/jma/indexe.html) from 2000 to 2015, the climate is characterized by relatively high temperatures and evenly distributed precipitation throughout the year. The average temperature for the year in this area is around 16.2 °C. The average maximum temperature and average minimum temperature is around 25.1 and 7.7 °C, respectively. The average precipitation for the year in this prefecture is around 1609.7 mm.

We conducted our research onto two experimental phases. In the first phase, we applied cover cropping-soybean rotations without P-application. In the second phase, we applied the same cover crop-soybean rotation with or without P-application. The first cover crop experiment (2007–2012) comprised three winter cover crop treatments such as winter wheat (Triticum aestivum L.), red clover (Trifolium pratense L.), oilseed rape (Brassica napus L.) and fallow (Fig. 1). These three cover crop treatments are used to cover the soil surface during winter and the fallow period in annual cropping systems in temperate regions of Japan. They have also been used in rotation with summer crops, not only to promote soil biological/chemical fertility, but also to modify the physical properties of soil and to suppress weeds. There were three replicate plots per treatment arranged in a randomized complete block design. Each plot had an area of 9 m2 (4.5 × 2 m). In this first phase, the experimental did not receive P fertilizers for over a five-year period. In the field plots used for the experiments, soybean (Glycine max (L.) Merr., cv: Enrei) had been cultivated to standardize soil biochemical conditions before the field trial started. As a preliminary investigation of soil chemical characteristics (0–15 cm soil depth) at this experimental site in 2014 before the study of phase two (Fig. 1), the soil pH ranged from 6.0 to 6.1 and total organic carbon (C) was 5.6–6.5%, respectively. Total nitrogen (N) and nitrate nitrogen content ranged from 0.41% to 0.48% and from 6.0 to 15.9 mg kg−1, respectively. Phosphate absorption coefficient ranged from 2,320 to 2,660. Further management details about the general information of the cover crop rotational system, seeding and sampling are presented in Higo et al. (2014).

Figure 1 Summary of the experimental design of this cover crop rotational study.

In phase two of our experiment, the same three cover crops from the first experimental phase (wheat, red clover and oilseed rape) were sown in rows, with spacing of 30 cm, in the cropped treatment on November 9, 2013, and November 18, 2014 except for red clover which was sown on March 16, 2015. Winter wheat (cv: Bandowase, mycorrhizal crop) seeds were sown at 200 kg ha−1 with N (ammonium sulfate) and K (potassium chloride) application rates of 100 and 90 kg ha−1, respectively. Oilseed rape seeds (cv: Michinokunatane, non-mycorrhizal crop) were sown at 30 kg ha−1 with N and K application rates of 100 and 50 kg ha−1, respectively. Red clover seeds (cv: Makimidori, mycorrhizal crop) were sown at 30 kg ha−1 with N and K application rates of 30 and 50 kg ha−1 in 2013 and 2015. The tops of the cover crops were cut close to the ground and removed on June 3, 2014, and June 16, 2015. In fallow, weeds were manually removed during the winter period.

We used a split plot design to divide the 4.5 × 2 m of the cover crop experimental plots into 2.25 × 2 m plots for the two P treatment plots (no P-application and P-application) of the phase two experiment (Fig. 1). Then, both no P and P-application plots were replicated three times in 2.25 × 2 m plots. The soybean (cv: Enrei) seeds were sown at a spacing of 60 × 15 cm on June 17, 2014, and June 17, 2015. To obtain the soybean grain yield, soybeans in each treatment were collected at the maturity stage (R8 growth stage) in early to late October in both years. In 2014 and 2015, the N and K application rates were 30 and 50 kg ha−1, respectively. In 2014, the amount of P (triple superphosphate) in the P-application plots was applied at 52.5 kg ha−1 as a sub-optimal level. The P-application did not increase the available soil P in 2014 because of the high P absorption coefficient (around 2,600). In 2015, the amount of P in the P-application plots was applied at 157.5 kg ha−1 to overcome the high P absorption fixation into the soil it as a P regime for comparing with soybean growth performance with no P- and P-application. The amount of P-application was moderate and optimal to increase the available soil P in this study. The content of available soil P (Truog P) was measured according to Truog (1930).

Soil and root sampling and root staining

The soil samples were randomly taken from 10 points in each replicate and pooled to one composite sample on June 17, 2014, and June 17, 2015, respectively. Soybean root samples were taken at the full bloom stage (R2 growth stage) on July 31, 2014 (44 days after planting), and August 6, 2015 (50 days after planting). The full bloom stage corresponds to the stage when the mycorrhizal colonization of soybean roots is usually at its highest (Zhang et al., 1995). In each rotation, the root samples were randomly collected from 10 plants (to a depth of 15 cm, the diameter of 20 cm) per replicate. The root samples were collected from the soil sample and maintained at −80 °C for DNA extraction and measurement of AMF colonization. The root samples were stained with a 5% (w/v) black ink-vinegar solution (Vierheilig et al., 1998), and the AMF colonization in the soybean roots was measured as described by Giovannetti & Mosse (1980).

Analysis of plant P and measurement of soybean grain yield

The aboveground plant parts of the 10 soybean plants were cut close to the ground at the full bloom stage and were randomly sampled on July 31, 2014, and August 6, 2015. To obtain the soybean grain yield, 10 soybean samples per plot in each treatment were collected at maturity stage in early to late October in both years. The aboveground soybean plant biomass and plant length were measured in all plots. The aboveground plant biomass and P uptake by soybeans were determined after the samples were oven dried at 80 °C for 48 h. The P uptake was determined using the molybdenum yellow colorimetric method (Murphy & Riley, 1962).

DNA extraction and nested polymerase chain reaction

Total genomic DNA was extracted from 150 mg of fresh root samples using the DNeasy Plant Mini Kit (Qiagen, Hilden, Germany) according to the manufacturer’s instructions. The genomic DNA pellet was stored at −30 °C until use in the nested polymerase chain reaction (PCR). The fragments in the fungal small subunit ribosomal DNA (SSU rDNA) were amplified using nested PCR method (Liang et al., 2008). The universal eukaryotic primer NS31 (forward) (5′-TTGGAGGGCAAGTCTGGTGCC-3′) (Simon, Lalonde & Bruns, 1992) and the fungus-specific primer AM1 (reverse) (5′-GTTTCCCGTAAGGCGCCGAA-3′) (Helgason et al., 1998) were used in the first PCR to amplify the 5′ end of the SSU rDNA region for comprehensive taxon sampling for the Glomeromycota (Schüßler et al., 2001a; Schüßler, Schwarzott & Walker, 2001b). Three subsamples per plot were amplified in a 20 μL reaction mixture containing 2 μL of 10-fold genomic DNA (around 1 to 5 ng/μL), 0.2 μM of each primer and 2× GoTaq Green Master Mix (Promega, Madison, WI, USA) using a Mastercycler ep Gradient (Eppendorf, Hamburg, Germany). The PCR condition was composed of initial treatment at 94 °C for 1 min; 30 cycles at 94 °C for 1 min, 66 °C for 1 min and 72 °C for 90 s; and a final extension at 72 °C for 10 min. The first PCR products were diluted 10-fold and used as templates for the second PCR using the nested primers NS31-GC (forward) (5′-CGCCCGGGGCGCGCCCCGGGCGGGGCGGGGGCACGGGGGTTGGAGGGCAAGTCTGGTGCC-3′) (Kowalchuk, de Souza & van Veen, 2002) and Glo1 (reverse) (5′-GCCTGCTTTAAACACTCTA-3′) (Cornejo et al., 2004). Three subsamples per plot were amplified in a 20 μL reaction mixture containing 2 μL of 10-fold 1st PCR amplicons, 0.2 μM of each primer and 2× GoTaq Green Master Mix (Promega, Madison, WI, USA) using a Mastercycler ep Gradient (Eppendorf, Hamburg, Germany). The PCR protocol was composed of initial treatment at 95 °C for 5 min; 35 cycles at 94 °C for 45 s, 52 °C for 45 s and 72 °C for 1 min; and a final extension at 72 °C for 30 min. Gel electrophoresis separated amplification products on 1% agarose gel, and the approximately 250 bp DNA amplicons were visualized by staining with ethidium bromide.

PCR-denaturing gradient gel electrophoresis

Three independent PCR products were pooled together, and then 20 μL of the nested PCR product was subsequently analyzed by denaturing gradient gel electrophoresis (DGGE) on a DCode Universal Mutation Detection System (Bio-Rad Laboratories, Piscataway, NJ, USA). Standard DNA markers were created by individually PCR-amplifying DNA extracted from root samples by Higo et al. (2015b). The PCR-DGGE condition was based on the method of Higo et al. (2015b). The gels containing 6.5% acrylamide were poured with a gradient of 35–55% denaturant. All DGGE analyses were performed in a 1× TAE buffer at a constant temperature of 55 °C at 50 V for 60 min, followed by 50 V for 960 min. The gels were stained with SYBR Green diluted in 1× TAE buffer (1:10,000) for 20 min, UV illuminated and digitally photographed (Figs. S1 and S2). Pictures were digitized by Phoretix 1D Pro (Nonlinear Dynamics Ltd., Newcastle upon Tyne, UK). We calculated Shannon index (H′) from these data, expressed by the number of DGGE bands in each root sample. Fromin et al. (2002) and Schneider et al. (2015) mentioned that visual observation of the DGGE gel revealed the presence of multiple bands in all samples (a band represents a distinct taxon in theory).

Quantification of specific root AMF taxa using a quantitative real-time PCR

The abundance of six-selected typical AMF taxa including R. irregularis, F. mosseae, Claroideoglomus claroideum, Gigaspora margarita, Cetraspora pellucida and Diversispora celata was measured using quantitative real-time PCR (qPCR) with taxon-specific primers and hydrolysis (TaqMan) probes targeting large subunit ribosomal DNA (LSU rDNA) genes (Wagg et al., 2011; Thonar, Erb & Jansa, 2012). The partial LSU rDNA genes of R. irregularis, F. mosseae, C. claroideum, G. margarita and Ce. pellucida followed the method described by Thonar, Erb & Jansa (2012). We also used the method described by Wagg et al. (2011) to quantify D. celata. Each PCR sample contained a total volume of 10 μL that consisted of 2 μL water, 400 nM each of forward primer and reverse primer, 100 nM TaqMan probe and 2× FastStart TaqMan Probe Master Mix + 2 μL of 10-diluted genomic DNA. The qPCR was carried out using a LightCycler 96 (Roche Diagnostics, Rotkreuz, Switzerland). The qPCR cycling conditions were as follows: initial denaturation at 95 °C for 15 min, followed by 45 cycles with denaturation at 95 °C for 10 s and annealing at the optimized temperature for each primer/probe combination for 30 s and elongation at 72 °C for 1 s.

Statistical analysis

We used an arcsine-square root transformation to normalize the data of AMF colonization in the soybean. The available soil P, growth parameters and AMF diversity data were transformed using a natural logarithm. The abundance of AMF tax was log (x + 1) transformed to reduce heteroscedasticity in the data. First, a generalized linear model was used to determine the effects of P-application and cover cropping and their interactions on each parameter in this study of a split plot design in R 3.3.2. Next, differences among means, where analysis of variance was significant, were assessed using Tukey’s honestly significant difference test (P  <  0.05) using the multcomp package in R 3.3.2. Data for the significance of differences between P-application treatments among cover crop systems were assessed using Student’s t-test.

A permutational multivariate analysis of variance (PERMANOVA) was performed using the vegan package in R to investigate the effect of P-application and cover crop systems on AMF community structure (Hammer, Harper & Ryan, 2001). To analyze the relationship of cover cropping and P-application with respect to AMF community structures (AMF communities), a redundancy analysis (RDA) (gradient length <4) was performed as the multivariate analysis using the vegan package in R 3.3.2. The presence/absence data matrix was composed of the abundance of DGGE bands and cover crop management or P-application. The environmental variable of cover cropping and P-application was coded as a dummy variable (0 and 1). Goodness-of-fit statistics (R2) of measured factors fitted to the RDA ordination of the AMF community were calculated using the envfit function in the vegan package with P-values based on 999 permutations (Oksanen, 2017). To investigate if AMF community structure differed significantly between P-application or cover crop management, the PERMANOVA was performed with 999 permutations using the adonis function in the vegan package in R.

The network graph included the correlation coefficients between soybean growth performance and AMF parameters using the igraph package in R, and then the network graph was described using Cytoscape for visualizing complex networks (www.cytoscape.org/). In this model, the AMF taxa abundance was represented by the scores of the first component of the principal component analysis (PCA) in this study. Pearson’s correlation coefficient was expressed as the indication of the strength of the connections.

Results

Available soil P and AMF colonization

In this study, the P-application in 2014 did not change the available soil P regardless of cover crop systems, whereas a significant difference was found in the available soil P in all of the cover crop systems compared with the no P-application plots in 2015 (Fig. 2A).

Figure 2 Impact of cover cropping and phosphorus (P) regime on the available soil P and root colonization of arbuscular mycorrhizal fungi (AMF) in the soybean at full bloom stage (R2) in 2014 and 2015.

Different letters within the same column for each variable in no P- or P-application plot among the cover crop systems show a significant difference by Tukey’s test at the 5% level. n.s. = not significant, * and ** indicates a significant difference at 5% and 1% level by t-test. Error bars indicate standard errors of the mean (n = 3). (A) Available soil P, (B) AMF colonization.

Overall, the AMF colonization in the soybean regardless of P-application and cover crop systems was never greater than 20% (Fig. 2B). In the no P-application plots, cover cropping affected the AMF colonization at the R2 stage in 2014 and 2015 (Fig. 2B). In 2014 and 2015, the AMF colonization under wheat and red clover treatments with no P-application plots tended to be higher than that of oilseed rape and fallow with no P-application plots. Contrary to the results of the no P-application plots, the AMF colonization in all of the cover cropping treatments with P-application was at a similar level. This similar tendency in AMF colonization with regard to cover cropping was observed between 2014 and 2015. Overall, our results showed that the cover cropping contributed to the AMF colonization in the soybean roots only for no P-application plots, whereas P-application eliminated differences in AMF colonization due to cover cropping.

Plant growth, P uptake and grain yield

According to Fig. 3A, the aboveground plant biomass in soybeans at the R2 stage did not vary among cover crop systems in both 2014 and 2015. In 2014 and 2015, the aboveground plant biomass under wheat and fallow treatments with no P-application plots tended to be higher than that of red clover and oilseed rape treatments with no P-application plots. In 2015, but not 2014, P-application plots, the aboveground plant biomass of soybeans was more than double than those of the no P-application plots. The aboveground biomass for the P- and no P-application plots were significantly different for red clover (2015), oilseed rape (2014 and 2015) and fallow (2014 and 2015).

Figure 3 Impact of cover cropping and phosphorus (P) regime on the growth performance of the soybean at full bloom stage (R2) and maturity stage (R8) in 2014 and 2015.

Different letters within the same column for each variable in no P- or P-application plot among the cover crop systems show a significant difference by Tukey’s test at the 5% level. n.s. = not significant, * and ** indicates a significant difference at 5% and 1% level by t-test. Error bars indicate standard errors of the mean (n = 3). (A) Aboveground biomass, (B) Plant P uptake, (C) Grain yield.

Our results as shown in Fig. 3B revealed that cover cropping did not have a significant effect on the plant P uptake of soybeans regardless of the P-application plots. However, the plant P uptake in soybeans was significantly increased by the P-application in 2014 and 2015. Moreover, there was a significant difference in the plant P uptake between the P- and no P-application plots for fallow in 2014 and 2015. In 2014 and 2015, the plant P uptake under wheat and fallow treatments with no P-application plots tended to be higher than that of red clover and oilseed rape with no P-application plots. Contrary to the results of the no P-application plots, the plant P uptake in all of the cover cropping treatments with P-application was inconsistent between 2014 and 2015. The plant P uptake in oilseed rape and fallow treatments was higher than compared with wheat and red clover treatments in 2014. However, in 2015, the plant P uptake was at a similar level in wheat, oilseed rape and fallow, but the plant P uptake in red clover treatment was lowest throughout the two-year study.

We found that the grain yield in soybeans was influenced by the cover crop systems only in 2014 (Fig. 3C). The grain yield in fallow treatment was highest among the cover crop treatments without P-application in both years. The P-application had a significant effect on the grain yield of soybean in both 2014 and 2015. The soybean grain yields at the P-application plots in the experiment were more than double in both 2014 and 2015. We also found that there were significant differences in the grain yield between the P- and no P-application plots for red clover (2014 and 2015) and oilseed rape (2015) managements.

Diversity of AMF communities and taxa abundance in the roots of soybean

The diversity index (H′) in soybeans at the R2 stage was significantly influenced by P-application and cover cropping in both 2014 and 2015 (Fig. 4). The H′ in oilseed rape and fallow treatments regardless of P-application and cover cropping was higher than compared with wheat and red clover treatments, although oilseed rape and fallow are non-mycorrhizal cover cropping. This tendency was opposite to that found for the AMF colonization of soybeans. Additionally, P-application stimulated H′ in all cover cropping treatments in both years. The H′ in the P-application plots regardless of cover crop treatments was higher than that of the no P-application plots.

Figure 4 Impact of cover cropping and phosphorus (P) regime on the diversity of AMF communities colonizing the soybean roots at full bloom stage (R2) in 2014 and 2015.

Different letters within the same column for each variable in no P- or P-application plot among the cover crop systems show a significant difference by Tukey’s test at the 5% level. n.s. = not significant, * and ** indicates a significant difference at 5% and 1% level by t-test. Error bars indicate standard errors of the mean (n = 3).

Our results showed that the six-selected AMF taxa were not influenced by cover cropping in 2014 (Table 1). However, the P-application had a significant effect on the abundance of all six-selected AMF taxa in 2014. The abundance of five AMF taxa in the P-application plots tended to be higher than those of the no P-application plots regardless of the cover crop systems. By contrast, in 2015, the abundance of R. irregularis in the P-application plots significantly decreased compared with that in the no P-application plots for wheat, red clover, and oilseed rape. Ce. pellucida also decreased but only in oilseed rape treatment. The abundance of the other AMF taxa was not affected by P-application. As in 2014, there was no significant effect of cover cropping on any taxa.

Table 1 Impact of cover cropping and phosphorus (P) regime on the abundance of different arbuscular mycorrhizal fungal (AMF) taxa colonizing soybean roots at full bloom stage (R2) in 2014 and 2015.

Rotation year	P-application	Cover crop	R. irregularis (Log gene copies mg−1dry root)	F. mosseae (Log gene copies mg−1dry root)	C. claroideum (Log gene copies mg−1dry root)	
No P-application	P-application		No P-application	P-application		No P-application	P-application		
2014	52.5 kg ha−1	Wheat	15.4 (0.89) a	18.0 (0.28) a	n.s.	9.8 (0.63) a	10.8 (0.28) a	n.s.	17.0 (0.44) a	17.5 (0.16) a	n.s.	
Red clover	16.5 (0.10) a	18.0 (0.58) a	n.s.	9.2 (0.26) a	11.6 (0.65) a	*	16.3 (0.36) a	18.6 (0.50) a	*	
Oilseed rape	16.9 (0.74) a	19.0 (0.05) a	*	8.5 (0.48) a	11.9 (0.37) a	**	15.4 (0.37) a	19.0 (0.63) a	**	
Fallow	15.6 (0.44) a	17.6 (0.20) a	*	9.2 (0.61) a	10.7 (0.17) a	n.s.	16.1 (0.65) a	17.8 (0.22) a	n.s.	
Analysis of variance										
P-application (A)	P < 0.001			P < 0.001			P < 0.001			
Cover cropping (B)	n.s.			n.s.			n.s.			
A×B	n.s.			n.s.			P < 0.05			
2015	157.5 kg ha−1	Wheat	10.4 (0.32) a	9.3 (0.21) a	*	9.6 (0.33) a	9.7 (0.06) a	n.s.	13.1 (0.41) a	13.7 (0.00) a	n.s.	
Red clover	10.7 (0.27) a	9.5 (0.30) a	*	10.0 (0.29) a	10.3 (0.07) a	n.s.	13.7 (0.32) a	13.9 (0.23) a	n.s.	
Oilseed rape	10.6 (0.12) a	9.5 (0.26) a	*	10.1 (0.12) a	10.2 (0.10) a	n.s.	13.7 (0.15) a	14.0 (0.20) a	n.s.	
Fallow	10.1 (0.24) a	9.6 (0.28) a	n.s.	9.7 (0.17) a	10.2 (0.37) a	n.s.	13.3 (0.14) a	14.1 (0.66) a	n.s.	
Analysis of variance										
P-application (A)	P < 0.001			n.s.			n.s.			
Cover cropping (B)	n.s.			n.s.			n.s.			
A×B	n.s.			n.s.			n.s.			
Rotation year	P-application	Cover crop	G. margarita (Log gene copies mg−1dry root)	Ce. pellucida (Log gene copies mg−1dry root)	D. celata (Log gene copies mg−1dry root)	
No P-application	P-application		No P-application	P-application		No P-application	P-application		
2014	52.5 kg ha−1	Wheat	14.0 (0.77) a	15.6 (0.31) a	n.s.	13.9 (0.55) a	14.9 (0.23) a	n.s.	16.1 (0.70) a	17.2 (0.17) a	n.s.	
Red clover	13.6 (0.38) a	14.8 (0.18) a	*	13.4 (0.11) a	16.1 (0.51) a	**	15.6 (0.15) a	18.5 (0.53) a	**	
Oilseed rape	13.0 (0.86) a	15.1 (0.35) a	n.s.	12.6 (0.63) a	15.9 (0.16) a	**	15.2 (0.59) a	17.7 (0.25) a	*	
Fallow	13.3 (0.64) a	15.5 (0.35) a	*	13.1 (0.35) a	14.9 (0.14) a	**	15.6 (0.49) a	17.4 (0.17) a	*	
Analysis of variance										
P-application (A)	P < 0.001			P < 0.001			P < 0.001			
Cover cropping (B)	n.s.			n.s.			n.s.			
A×B	n.s.			P < 0.05			n.s.			
2015	157.5 kg ha−1	Wheat	6.4 (0.51) a	6.4 (0.36) a	n.s.	14.1 (0.43) a	13.7 (0.12) a	n.s.	14.4 (0.46) a	13.9 (0.26) a	n.s.	
Red clover	6.9 (0.56) a	7.0 (0.21) a	n.s.	14.5 (0.22) a	13.9 (0.40) a	n.s.	14.6 (0.35) a	14.1 (0.48) a	n.s.	
Oilseed rape	6.4 (0.43) a	6.5 (0.54) a	n.s.	14.9 (0.21) a	13.6 (0.19) a	*	14.6 (0.15) a	14.5 (0.29) a	n.s.	
Fallow	5.9 (0.21) a	6.3 (0.51) a	n.s.	14.2 (0.08) a	13.9 (0.24) a	n.s.	14.7 (0.54) a	14.1 (0.35) a	n.s.	
Analysis of variance										
P-application (A)	n.s.			P < 0.01			n.s.			
Cover cropping (B)	n.s.			n.s.			n.s.			
A×B	n.s.			n.s.			n.s.			
Note:

Different letters within the same column for each variable in no P- or P-application plot among the cover crop systems show a significant difference by Tukey’s test at the 5% level. n.s. = not significant, * and ** indicates a significant difference at 5% and 1% level by t-test. Numbers denote means of n = 3 and standard error in parentheses.

Relationships among AMF communities, cover cropping and P-application

We used an RDA to identify the relationships among AMF communities in soybean roots with cover crop management and P-application (Figs. 5A and 5B). In 2014 and 2015, the RDA trends clearly showed that the P-application noticeably altered the AMF community structure in the soybean roots. In 2014, the ordination diagram indicates that oilseed rape (R2 = 0.756, P = 0.001) contributed significantly to the variation in AMF root communities (Fig. 5A). However, wheat (R2 = 0.095, P = 0.349), red clover (R2 = 0.138, P = 0.191) and fallow (R2 = 0.040, P = 0.630) did not contribute to the variation in the AMF root communities. Additionally, the P-application treatment (R2 = 0.801, P = 0.001) and no P-application treatment (R2 = 0.801, P = 0.001) contributed to the variation in the AMF root communities. In 2015, the ordination diagram indicates that red clover (R2 = 0.704, P = 0.001) contributed significantly to the variation in the AMF root communities (Fig. 5B). However, wheat (R2 = 0.154, P = 0.181), oilseed rape (R2 = 0.129, P = 0.255) and fallow (R2 = 0.173, P = 0.141) did not contribute to the variation in the AMF root communities. Furthermore, the P-application treatment (R2 = 0.743, P = 0.001) and no P-application treatment (R2 = 0.743, P = 0.001) contributed to the variation in the AMF root communities. A PERMANOVA was also carried out to examine the relative importance of each agricultural management for the AMF root communities. The PERMANOVA showed that P-application significantly affected the AMF root community structure (2014: F = 4.263, P = 0.001, 2015: F = 4.226, P = 0.001), but cover crop management did not impact the AMF root communities (2014: F = 1.193, P = 0.189, 2015: F = 1.669, P = 0.057).

Figure 5 Redundancy analysis (RDA) biplot showing the relationship among the AMF communities, cover cropping and phosphorus (P) application in 2014 (A) and 2015 (B).

The eigen values of the first and second axes were 3.680 and 2.253, respectively. Solid lines indicate significant effects, and dashed lines indicate non-significant effects. The environmental variable of cover cropping and P-application was coded as a dummy variable (0 and 1). The no-P and P-application are distinguished by clear shapes for no-P application and filled shapes for P-application plots. Wheat = circle, red clover = square, oilseed rape = triangle, fallow = diamond.

Response of soybean growth to AMF parameters

In the soybean growth response, the relationships between available soil P and soybean growth performance was not linear in the cropping system with no P-application (Fig. 6A–6C). The difference in the soybean growth performance was small with no P-application. The relationships between available soil P and soybean growth performance such as plant biomass (r = 0.874), plant P uptake (r = 0.821) and grain yield (r = 0.801) was significantly linear in the cropping system with P-application. With the AMF contributions to soybean growth performance, the relationships between AMF colonization and soybean growth was not linear in the cropping system with and without P-application (Fig. 6D–6F). The P-application significantly improved the linear relationships between the diversity index and soybean growth performance (Fig. 6G–6I). The relationships between the diversity index and soybean growth performance such as plant biomass (r = 0.967), plant P uptake (r = 0.967) and grain yield (r = 0.928) was positively correlated in the cropping system with P-application.

Figure 6 Soybean growth and arbuscular mycorrhizal fungal (AMF) response for phosphorus (P) nutrition, growth and grain yield with or without P-application in this study.

The observations with and without P-application are denoted by open square and open circles, respectively. The number above each line represents the value of the Pearson’s correlation coefficient. Solid lines indicate P-application plots, and dashed lines indicate no P-application plots. The data of both years in 2014 and 2015 are included. A, B and C indicates available soil P vs Plant biomass, Plant P uptake and Grain yield, respectively. D, E and F indicates AMF colonization vs Plant biomass, Plant P uptake and Grain yield, respectively. G, H and I indicates diversity index vs Plant biomass, Plant P uptake and Grain yield, respectively.

To understand the role of AMF parameters on soybean performance and how they link to the available soil P and soybean growth, we used a network analysis to identify the relationships between AMF parameters in soybean roots and soybean growth in this study (Fig. 7). The results showed the same tendency with the linear analysis in the two-year experiment. The relationships between the diversity index and available soil P were related to the soybean growth performance such as plant P uptake, plant biomass and grain yield. However, each AMF taxa abundance and AMF colonization were not related to the soybean growth responses, especially grain yield, throughout the experiment.

Figure 7 Network analysis showing relatedness between soybean performance and arbuscular mycorrhizal fungal (AMF) parameters in the two-year field trial.

Each circle represents a variable in the model, while the number above each arrow represents the value of the Pearson’s correlation coefficient. Solid lines indicate positive relationships, and dashed lines indicate negative relationships. The data of both years in 2014 and 2015 are included. * and ** show a significant difference (P < 0.05 and P < 0.01, respectively).

Discussion

Impact of P-application and cover cropping on colonization

It is well known that cultivation of preceding crops or fallow as well as P-application impacts AMF colonization of subsequent crops (Karasawa, Kasahara & Takebe, 2002; Karasawa & Takebe, 2012; Isobe et al., 2014). Mycorrhizal cover crops or oilseed rape slightly increased AMF colonization of subsequent soybean compared with fallow as a control (Fig. 2B), in agreement with previous studies (Karasawa, Kasahara & Takebe, 2002; Karasawa & Takebe, 2012; Isobe et al., 2014). Some studies also reported that Brassicaceae plants decreased colonization in subsequent crops during just early growth stages compared with mycorrhizal crops (Gavito & Miller, 1998; Sorensen, Larsen & Jakobsen, 2005). Thus, the cultivation of oilseed rape in a rotation with soybean during the winter period may not have necessarily interfered with AMF colonization of subsequent soybean in this study.

In addition, AMF colonization is inhibited under high P-application (Kahiluoto et al., 2001; Balzergue et al., 2011). Plants can fail to react to AMF when available soil P is extremely low (Ryan et al., 2002). Miranda & Harris (1994) also reported that deficiency of available soil P inhibited AMF colonization. On the contrary, Gosling et al. (2013) indicated that there was no significant decrease in AMF colonization in soybeans under high P availability in soil. Plants can control AMF colonization depending on their nutritional status (Smith & Read, 2008) as well as under high soil P conditions. Bolan, Robson & Barrow (1984) also reported that a moderate amount of P-application in P-limited soils might increase AMF colonization and benefits such as P availability for crop growth performance. Our results indicated that there was no effect of cover cropping on AMF colonization in soybeans under P-application, but cover cropping only increased AMF colonization when P was not applied (Fig. 2B). Thus, one possible reason for higher AMF colonization in P-application may be that the P-application may possibly stimulate potential AMF activities such as hyphal growth to colonize soybean roots by improving P fertility that prefers AMF in the soil.

Impact of P-application and cover cropping on the diversity of root AMF communities

Previous studies have reported that P fertilization had no significant effect on the diversity of AMF in maize roots and its rhizosphere soils under a long-term field experiment (Liu et al., 2016; Wang, White & Li, 2017). On the contrary, Lin et al. (2012) found that chemical fertilizers decreased AMF diversity in a long-term field experiment. Moreover, Gosling et al. (2013) reported that the AMF community diversity in soybean roots decreased due to the high availability of soil P in a field study. In addition, plants can directly gain enough nutrients from the soil in a nutrient-rich environment without benefit from AMF. As a result, the diversity of AMF communities can also decrease (Liu et al., 2015). Surprisingly, our results indicated that the diversity of AMF communities in soybeans, regardless of cover crop management, tended to increase as a result of P-application (Fig. 4), in disagreement with our hypothesis. Also, the shift of AMF communities was obvious from the results of RDA trends that showed that the P-application changed the AMF community structure in the soybean roots rather than the cover crop systems (Fig. 5). Wakelin et al. (2012) and Maček et al. (2011) implied that abiotic selective pressures such as soil fertility determine the AMF community structure. The observed increase in AMF diversity as a result of P-application can be linked to the degree of selective pressure for mycorrhization in soybean roots. Some P-unresponsive taxa may have been dominant in the experimental field under the cover crop rotational system. Increasing the available soil P can decrease the selective pressure, and this could increase the opportunity for P-responsive species to establish soybean roots. Thus, one possible explanation for this result was that the activity of AMF could have been inhibited due to soil P depletion of the P-unfertilized five-year continuous crop rotational system. This could be one reason why P-application increased the AMF diversity of soybean crops.

Furthermore, we found that cover cropping did not impact the AMF root communities in soybeans from the result of PERMANOVA (Fig. 5). The P-application eliminated the effect of cover cropping in both AMF colonization and diversity in the roots of subsequent soybeans. Turrini et al. (2016) and Higo et al. (2018) indicated that a shift in indigenous AMF communities in the subsequent maize roots was independent of cover crop identity and diversity. Higo et al. (2014) also found that cover crop rotations did not impact AMF communities in the roots of subsequent soybean. However, rotation year affected the AMF communities in soybean roots suggesting that climate or other environmental conditions were more imperative than cover crop management. Therefore, the P-application may have influences on AMF communities in soybean roots, suggesting that fertilizer application or other factors such as soil chemical properties and other environmental factors can be more important than cover cropping. This was the case of R. irregularis in the soybean as our study found that its abundance was influenced by P-application, but not by cover cropping in this study.

Impact of P-application and cover cropping on the abundance of root AMF taxa

Previous studies have reported that AMF have different niches and are well known to prefer to inhabit different soils (Johnson, 1993). Moreover, fertilization may directly favor species that grow better in enriched soils (Dumbrell et al., 2010). Wakelin et al. (2012) have reported that R. irregularis decreased as a component of the AMF communities with increasing available soil P, in agreement with our study (Table 1). The fluctuation in abundance of AMF taxa as a result of P-application could link to the preference of fertilization or inhabiting soil conditions among AMF in soybean roots. R. irregularis has been observed in a different type of lands, and can have an ability of high tolerance for environmental factors due to a strategy of life history as a generalist (Börstler et al., 2008). Thus, it is likely that the R. irregularis may tolerate under the low-P soil conditions in this cover crop rotational system.

Impact of P-application and cover cropping on the soybean performance

According to a study by Jansa, Smith & Smith (2008), the growth of Allium porrum with three inoculated AMF species (F. mosseae, C. claroideum and R. irregularis) was enhanced compared to that of A. porrum when each AMF species was mono-inoculated. Similarly, in the results of network analysis and growth response of soybean to AMF parameters, we found that the aboveground plant P and biomass of soybeans during the R2 stage and the grain yield of soybeans were positively correlated with the AMF diversity in the roots of soybeans with P-application (Figs. 6 and 7). Gosling, Jones & Bending (2016) also reported that increased benefit from high AMF diversity on the growth of Allium cepa was found compared to mono-inoculated. On the other hand, no positive correlations were observed between AMF colonization and soybean grain yield regardless of P-application (Figs. 6 and 7), in disagreement with a previous study (Isobe et al., 2014). One possible explanation may be that a moderate amount of P-application in P-limited soils may increase root colonization and benefits such as P availability for crop growth performance (Bolan, Robson & Barrow, 1984). The increase of AMF colonization with P-application was only slight (i.e., under 20%), and the increase may be related to the direct response of soybeans to increasing P availability that might obscure the response according to the increased AMF colonization. Furthermore, the AMF communities were assessed by only flowering seasons and crop phenology may be important for the shift of AMF communities. We simply did not take any samples at different crop growth stages and future studies examining according to different crop phenology would be needed to better understand the relationships between crop phenology and AMF diversity with P-application.

Furthermore, the introduction of cover crops can increase the amount of carbon, such as organic matter, to serve as an energy source for biological activity (Jokela et al., 2009). In this study, our cover crop systems did not improve growth performance such as plant biomass and P uptake of soybean at the full bloom stage (Figs. 3 and 7). In fact, cover cropping decreased grain yield of soybeans without P-application in 2014 compared with fallow, whereas the P-application enhanced the growth and yield of soybeans. There is one possible reason why cover cropping did not improve soybean growth performance. That may be due to the continuous nutritional removal in the five-year P-unfertilized crop rotational system because the top dry matters of cover crops were not incorporated into the soil. Therefore, further investigation into the relationships among AMF diversity, P-application and cover cropping on soybean growth performance would be required to gain more benefit from AMF in cover crop rotational systems.

Conclusion

The main conclusions from this experiment are that P-application was more important than cover cropping in AMF communities and soybean growth performance under P deficit conditions (in fact, P-application seems to eliminate the effect of cover cropping in both AMF colonization and diversity). In addition, P-application can have beneficial effects on the diversity of AMF communities in the P-unfertilized continuous crop rotational system. These differences in the AMF communities may relate to soybean productivity and P-use efficiency in cover crop rotational systems. Additionally, a higher diversity of AMF communities found in soybean roots with P-application can contribute to the potential for P uptake and growth under winter cover crop rotations with soybean. Thus, these results indicated that the soybean performance could be partially related to the interaction of P-application with AMF diversity. However, we still need to investigate how to improve agronomic benefits from AMF diversity associated with soybean plants, which will give useful information on appropriate P management and cover crop choices in cover crop rotational systems.

Supplemental Information

Supplemental Information 1 Impact of cover cropping and phosphorus (P) application on the PCR-DGGE band patterns of AMF communities colonizing soybean roots at full bloom stage (R2) in 2014.

1) M: DNA Marker. 2) Each of the numbers in the lanes shows the replicate in each cover crop treatment.

Click here for additional data file.

Supplemental Information 2 Impact of cover cropping and phosphorus (P) application on the PCR-DGGE band patterns of AMF communities colonizing soybean roots at full bloom stage (R2) in 2015.

1) M: DNA Marker. 2) Each of the numbers in the lanes shows the replicate in each cover crop treatment.

Click here for additional data file.

Supplemental Information 3 Statistical analysis.

Click here for additional data file.

Additional Information and Declarations

Competing Interests

Author Contributions

Data Availability

The authors declare that they have no competing interests.

Masao Higo conceived and designed the experiments, performed the experiments, analyzed the data, contributed reagents/materials/analysis tools, prepared figures and/or tables, authored or reviewed drafts of the paper, approved the final draft.

Ryohei Sato performed the experiments, analyzed the data, contributed reagents/materials/analysis tools.

Ayu Serizawa performed the experiments, analyzed the data, contributed reagents/materials/analysis tools.

Yuichi Takahashi performed the experiments, contributed reagents/materials/analysis tools, guidance on lab techniques.

Kento Gunji performed the experiments, contributed reagents/materials/analysis tools, guidance on lab techniques.

Yuya Tatewaki performed the experiments, contributed reagents/materials/analysis tools, guidance on lab techniques.

Katsunori Isobe conceived and designed the experiments, contributed reagents/materials/analysis tools, guidance on lab techniques, general advising.

The following information was supplied regarding data availability:

The raw data has been supplied as Supplemental Dataset Files.

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
