# Peer review of "Can phosphorus application and cover cropping alter arbuscular mycorrhizal fungal communities and soybean performance after a five-year phosphorus-unfertilized crop rotational system?"

_PeerJ, doi:10.7717/peerj.4606_

## Round 0.1 · original submission · Major Revisions

Please read the reviewers' comments carefully. There are several common themes:

1) The discussion was poorly written and repetitive with the results. Please rewrite to improve this.

2) The methods are poorly described and poorly written. I suggest a figure to convey the field design. The statistical analysis needs a more detailed description.

3) Several reviewers as well as myself note that your conclusions are often not justified based on the statistical analysis. For example in the section "Plant growth, P uptake and grain yield" you often describe which treatment is lowest e.g. "The aboveground plant biomass of soybeans was lowest with red clover"; however these differences are not significant and therefore the conclusion is invalid. The correct conclusion here is "cover crop did not have a statistically significant effect on aboveground plant biomass". I give this one example but this is a problem throughout.

In addition:

Please address the specific points raised by each reviewer.

Note that reviewer three provides an annotated PDF.

Also they found many mistakes in the citations. These can be avoided if you use a citation manager like Zotero (Free!) or endnote, and use web rather than manual entry. In any case fix the citations!

In spite of one reviewer's suggestion to only present 2015 I favor keeping both years data in the main results.

If you follow the suggestion to present the results as figures (which would make it easier to understand) please keep the table also.

You do not need to state the range of the observations, etc, in the results as this is clear from the tables.

In figure 3: I don't understand why there are arrows in the network diagram. Correlations are not directional so why are your connections directional? Remove the arrowheads. In figure 3 shouldn't you only retain connections where the correlation is significant?

Reviewer 1 ·

Basic reporting

The manuscript by Higo et al give an investigation concerning AMF community diversity and phosphorus application in soybean roots with different rotation systems. The authors found that P application and crop rotational systems changed the diversity and community composition of AMF and AMF diversity influenced positively plant biomass, plant P uptake and grain. Experimental settings have a certain degree of rationality, the author did a lot of work and the data is also very abundant. Literature references, sufficient field background/context were provided. I suggest that the data of 2014 can be placed in the supplementary information.

Experimental design

Original primary research within Aims and Scope of the journal. Research question well defined, relevant & meaningful. However, the Methods for describing Statistical analysis were not very detailed, it should be written more substantial.

Validity of the findings

The discussion is badly organized and is not real discussion. So much redundant information presented in the Result and Discussion. I am not convinced the data and statistics support the authors' hypotheses in the current version.

Additional comments

The manuscript by Higo et al give an investigation concerning AMF community diversity and phosphorus application in soybean roots with different rotation systems. The authors found that P application and crop rotational systems changed the diversity and community composition of AMF and AMF diversity influenced positively plant biomass, plant P uptake and grain. Experimental settings have a certain degree of rationality, the author did a lot of work and the data is also very abundant. However, this is not very novel. The impact of P on AMF community diversity and composition has been studied too much and the technological means of PCR-DGGE is relatively backward in microbial community research. I consider that there are many more advanced technical methods to do it. The Methods for Statistical analysis part was not very detailed, it should be written more substantial. In addition, the pictures are too hasty to publish, it would require repaint them better especially the last figure for network analysis.
This MS have two years of data in the text section so that it is difficult to catch the main idea. I suggest that the data of 2014 can be placed in the supplementary information and the author can discuss the data for 2015 in more detail, so the article will be more purposeful and convincing. The discussion is badly organized and is not real discussion. So much redundant information presented in the Result and Discussion. The discussion seems too superficial. I am not convinced the data and statistics support the authors' hypotheses.
All of the tables were hard to comprehend, it is better to changes these tables to figures.

It’s not necessary to present the data of 2014 in the body text, or you can consider showing it in the supplementary part.

Table 2: how do you link the aboveground plant biomass or other growth performance at R2 with the grain yield? It will be more scientific to characterize the differences of these growth performance from R2 to maturity stage, and analyze how the AMF root colonization affect P absorption and utilization during this stage.

Line 194: All test for tables were not provided. Please describe in detail.

Line 199 - 201: Please provide more details of RDA analysis and how did you define the all the environmental factors in Figure 1.

Line 280 – 286: How did identify these species of AMF? Please describe in detail.

Line 290: The legend describes of Figure 1, please change the “between” to “among”.

Line 290 - 293: The Monte-Carlo permutation test is used for whether the environmental factors explain the distribution of the community is significant. Please use PERMANOVA to test the AMF communities in different treatments.

Line 306 – 331: Different correlation lines should be represented by different types of segments and please provide more clear pictures.

Line 325 - 326: The Figure 3 was too hasty and should be repainted.

Line 352 - 357: You did not detect any functional traits, so, the discussion is a little bit desultory. Do these discussions make any sense to any of the conclusions?

Line 404: Remove “molecular”.

·

Basic reporting

Overall, I think that the manuscript could benefit from additional polishing to tighten the language and avoid redundancy, to ensure that the message being conveyed is clear and consistent throughout. Below are some examples I’ve highlighted throughout reading the manuscript:


The introduction does need to be expanded a little bit:
1) Lines 39-43: It would be good to mention the economic cost of fertilizer application over time. Do you have a reference that states that growers are applying less b/c of the cost? Also this first sentence does not clearly convey what your point: That the increasing costs is deterring growers from applying required levels of P containing fert. In these first few introductory sentences, it would also be good to get an idea of the environmental impact of P-containing fertilizer and why this necessitates a green revolution.
2) Lines 43-51: Can you give a couple examples of key AMF species and their benefits?
3) Lines 48-51: Redundant with the previous sentence.
4) Paragraph starting on line 53- at first pass, it is not clear that you are trying to give examples of management examples. Maybe re-work the first sentence?
5) Line 63-64: You need a citation that shows some stats / why Japanese growers are decreasing use of winter cover crops.
6) Line 67 and line 70: Please define what is meant by AMF composition and AMF diversity. It is not clear in how it is presented throughout the paper.
7) Paragraph starting on line 76: The first sentence is not clear- “In contrast, it is well known that a decrease occurs in AMF root colonization…” You can be clearer of what is intended by stating AMF root colonization is decreased following….
8) Line 78: “Specifically, Brassicaceae crops are not ..” This sentence is awkwardly worded. It makes it sound that the crop cannot establish a symbiotic relationship with itself.
9) Paragraph starting line 93: This paragraph is overall a bit redundant, consider revising. Also the stated hypothesis is a complicated in its wording.

Experimental design

In this section, the authors could further expand on the experimental set up. From what is written in lines 116-130. For example:
1) It is not clear at first that the authors replicated their experiment in 3 plots per P treatment. Also, it is not clear until the reader makes it to lines 144-157 that 10 individual replicated plants were included in each plot .
2) It would be good to know what accounts for the differences in P in the plots between the two sampled years. (If it can be explained).
3) Line 161- 162: Provide a reference for the molybdenum yellow colorimetric method.
4) Line 166-177:
a. “The DNA samples from the roots were used for the PCR templates after a 10-fold dilution”- what concentration was achieved?
b. Provide the primer sequences and thermocycler protocols for each set.
c. Can you please tell the reader why you chose to amplify the 18S SSU rDNA, AM1, NS31?
5) Lines 179-187: What DNA ladder was used?
6) Lines 189-192: Please provide the reader with the specific primer sets and thermocycler protocol.

Statistical analysis:
1) Lines 195-206:
a. Please clarify what the Monte Carlo permutation test was aiming to do. From this section and from the Results section, it is implied that the permutation is testing the significance of the quadrants from the RDA output. If that is true, then that should be described in this section and reiterated in the results section as well.
b. By looking at the tables reported within the manuscript, it appears that the statistics carried out were performed based on 3 replicates. Does this mean that the replicates within each plot (n=3), were averaged and this is what was used for the analysis described there?
c. What correlation test was performed for the data reported in figure 2 and 3? Spearman/Pearson?
d. How does the data from the two years correlate for plant measurements?

Validity of the findings

VALIDITY OF THE FINDINGS
1) The network model nicely illustrated in a figure the observations they had made and reported throughout. The most notable finding from this is the role that AM colonization and AM abundance has on Plant biomass and their relationship with Plant P uptake and available Soil P.

Additional comments

RESULTS:

1) Diversity of AMF communities and abundance:
a. Line 259: Please provide a reference for the use of DGGE bands for species richness. How is this measure useful / different or compare to just only using the Shannon index?
b. Paragraph starting on line 277: I am assuming that the AMF taxa was identified via qPCR as described in materials and methods. It is not clear from how this paragraph is started out that these taxa were identified via that method. Can you also elaborate as to why this method is a relevant measure to indicate the abundance of these species? Can you mention in the materials and methods how the samples were normalized? It seems like this method could bias the representation of the AMF taxa if the preparation method did not standardize the quantity of the cDNA.

2) Relatedness factors to AMF communities:
a. Please see my comment in the statistical analysis.
b. List the quadrants (I-IV).
c. List each RDA plot as A and B.

DISCUSSION:
1) Line 334: The stated hypothesis starting on line 93 does not match the reiterated hypothesis starting on this line.
2) Line 339: “However, other more important factors can be involved”, like what??
3) Line 366-369: Point is not clear here.
4) Line 389-391: What is meant by a net cost?


COMMENTS ON FIGURE LEGENDS:
Figure 1: Please divide up each RDA into quadrants, this will make it easier to discuss in the results section where you report the associated p-values from the Monte Carlo.

Figure 2: What correlation test was done? State that in legend. Is each correlation significant? That should be stated in the results.

Figure 3: How was each correlation value derived? Spearman/Pearson? State it in the legend.

Figure S1 and S2: It would be good if the lanes could be better labeled with their target names and also if the DNA ladder could be annotated.

GENERAL COMMENTS
I think the authors should prepare the reader for the caveats of the benefits of P and AMF in the introduction, so that once the reader gets to Figure 3 and reads through the discussion, it doesn’t seem like the findings reported here contradict what was known previously. Also, it would be good to mention this finding in the abstract, as I think it is key to the paper.

Reviewer 3 ·

Basic reporting

Revision of English language by a native is needed.
Manuscript should be condensed and guiding thread shoul be improved.
Introduction, discussion and conclusions are the weak points of the manuscript. They need a revision.

Experimental design

no comment

Validity of the findings

no comment

Additional comments

Abstract:
Only part of the results are mentioned. Discussion is poor.

Introduction need to be improved. Hypothesis should better support in introduction.
Guiding thread and connection with the rest of the document shoul be improved.

Material and methods: phases of the research shoul be better explained.

Results: they are interesting but the text is very repetitive, information is already shown in tables.

Discussion: main messages and guiding thread of each paragraph and, in general, of the whole discussion shoul be improved.
There are many and interesting results but some of them are poorly discussed or not discussed at all, i.e. the influence of cover cropping, and specifically the effect of the different species of cover crops (or fallow). In addition, the RDA analysis (Fig.1), or the impact on the abundance of different AMF taxa (and the meaning of that) (Table 4) are not discussed. In contrast, there is a paragraph poorly connected with results (Lines 382-395).

Conclusions: Conclusions need to be improved to better link to original research question and be limited to supporting results.

References:
Gilbert et al, 2000 instead of 2009
Njeru, instead of Njel
Schernier and Koide 1993 does not appear in list of references.
Kowalchuk et al, 2002 instead of 2001
Hicks and Loynachan 1987; does not appear in list of references.
Treseder 2002 instead of 2008
Liu et al 2015 instead of 2005 in list of references.
Alguacil et 2010 does not appear in the text
Higo et al. 2010 does not appear in the text

See comments on pdf.

Annotated reviews are not available for download in order to protect the identity of reviewers who chose to remain anonymous.

---

## Round 0.2 · Major Revisions

The reviewers all agree that the manuscript has been substantially improved, however there are still some changes needed. I am still rating this as a major revision because there are a number of contradictions in the manuscript (as noted by reviewer 3). Both reviewers 2 and 3 provide annotated files that I have reviewed and the provide excellent suggestions for you; please address the comments in those files.

Reviewer 1 ·

Basic reporting

Compared to the previous version, the latest manuscript has been greatly improved. The author earnestly answered all the questions that I had rasied. I do not think there is a big problem in the latest, except that the author should put all names of R software packages and functions into italics in the Statistical analysis section.

Experimental design

No problem.

Validity of the findings

No problem.

Additional comments

No problem.

·

Basic reporting

The manuscript still needs a fair bit of polishing. Please see commented manuscript attached.

Experimental design

The research phases are not well described. Please state why the experiment was broken into phases to better understand the flow.

Validity of the findings

no comment

Additional comments

Dear author,

Thank you for incorporating our suggestions. After careful reading, I think that the manuscript could still use a bit of polishing. In the next edits, redundancy can be further reduced and also pay attention to sentence and paragraph structure. I found some misspellings and typographical errors as well and have noted that in the track edits.

Reviewer 3 ·

Basic reporting

The new version was greatly improved compared with the original one, but it still needs changes for publishing.
Manuscript is sometime redundant and may be condensed a little more. Try to be more concise.
Despite suggestion of one of the reviewers, it is better to keep both years data in the main results. It seems strange that most figures (2-6) correspond to 2015 data but the last two figures (7 and 8) correspond to both year data. If 2014 data are kept as complementary information, please, justify this decision and provide some comments (not as detailed as with 2015).
Still explain better the two phases of experiment. It seems to be a split-plot design. The P fertilization is not clear in Phase 2.
Have you considered in the statistical analysis that it is a split-plot design?
Sometimes, results in the text are quite confusing and contradictory (mainly regarding figures 2-4). In some cases, the new text contradicts the original one. Text may be condensed in results.
Discussion is better in the new version but still needs some changes.
Indicate in figures that you used Tukey test.
Figures 5 and 7 are small and difficult to read.
Improve conclusions and remark the take-home message of the manuscript.
Revise references: Thonar et al, 2012 (2011) ; Wakelin et al 2012, Njeru et al 2014, Johnson et al, 1992, 2010a, 2010b.

Detailed comments in the manuscript.

Experimental design

no comment

Validity of the findings

no comment

Additional comments

no comment

Annotated reviews are not available for download in order to protect the identity of reviewers who chose to remain anonymous.

---

## Round 0.3 · Minor Revisions

Thank you for your revisions! This is getting close. I do ask you to address the minor revisions brought up by the two reviewers to improve readability and take this to publication-readiness.

·

Basic reporting

Basic reporting has improved since the last draft. Thank you for incorporating our suggestions. I did not find Figure descriptions for the supplemental figures. Also please indicate Marker sizes and also indicate which marker was used to compare amplicon sizes. Also, in each supplemental figure, there are 3 lanes per cover crop indication, however it is not clear whether each lane corresponds to a given AMF. Please have this indicated either in the figure or in the description.

In order to be clear to the audience, please remind your reader in the figure description that they are looking at data collected from 2 different P-application doses. It otherwise misleads the reader to think that the experiments carried out in the two years are fully repeated experiments.

I would also be interested in your interpretation of how a leguminous cover crop may impact soil nutrition and therefore your 2015 plant growth results. Did you perform any nutrient testing of the plots after each subsequent cover crop rotation?

I would like it if the author could label the quadrants of Figure 5 as they are described in text (i.e. quadrants I - IV).

Experimental design

Please state the developmental stage that the experiments were executed at. This information was provided in the results & figures descriptions, but not in the experimental design. Also, it should be stated why the researchers decided to apply a 3X Phosphorus application and also whether the researchers tested soil nutrition for 2015 as done in 2014 to account for differences in plant growth between the two years. Even the controls were larger in 2015. Was this increase hypothesis driven? Why was this not included as an experimental factor in the experimental design prior to initiating the studies?

Validity of the findings

Overall it seems like the authors did not find similar trends in the AMF population due to P application as was reported in previous literature. This is a point that is discussed, but I think that it would be good to get an understanding of how previous studies differ from the current one (i.e. growth chamber, greenhouse field experiments?) I raised this question in the discussion.

Additional comments

Overall, the manuscript has improved. There are some sentences that could benefit from clarification and help guide the reader as to why certain experiments/tests were performed. I made explicit comments/suggestions where further clarification is needed.

Reviewer 3 ·

Basic reporting

The new version was greatly improved. However, after so many punctual changes, the manuscript needs a global revision in order to be more concise and improve the style and readability of the text.

Please, see manuscript with comments.

Experimental design

no comment

Validity of the findings

no comment

Additional comments

no comment

Annotated reviews are not available for download in order to protect the identity of reviewers who chose to remain anonymous.

---

## Round 0.4 · Minor Revisions

Unfortunately this is still not ready for publication. I am not sure why it is taking so long to address the reviewers comments. I carefully read the manuscript and Reviewer 3's comments (hoping to eliminate some) but find that most really must be addressed before publication. In particular please address those labelled in the annotated manuscript at R2, R3, R6, R7, R13-22, R24-R27, R31, R37. I agree with the other ones as well so feel free to do more. But the ones above are critical. I'll evaluate the next version myself without sending to reviewers so hopefully we can get this wrapped up quickly.

·

Basic reporting

Improved basic reporting, addressed all concerns in previous revision.

Experimental design

No comment.

Validity of the findings

No comment.

Additional comments

Dear author,

Thank you for addressing the suggestions. The manuscript has improved greatly.

Best wishes!

Reviewer 3 ·

Basic reporting

The manuscript has been improved but still needs further improvements. It could be more condensed. Main messages should be clearly presented.

Experimental design

No comments

Validity of the findings

No comments

Additional comments

Please see comments in the document attached.
As I suggest to remove some studies, references were not checked.

Annotated reviews are not available for download in order to protect the identity of reviewers who chose to remain anonymous.

---

## Round 0.5 · accepted · Accept

Thanks for all of you work through the reivsions.

#